# Antibacterial, Antibiofilm and Anti-Inflammatory Activities of Eugenol Clove Essential Oil against Resistant *Helicobacter pylori*

**DOI:** 10.3390/molecules28062448

**Published:** 2023-03-07

**Authors:** Mahmoud K. M. Elbestawy, Gamal M. El-Sherbiny, Saad A. Moghannem

**Affiliations:** Botany and Microbiology Department, Faculty of Science, Al-Azhar University, Nasr City, Cairo 11884, Egypt

**Keywords:** eugenol, extraction, purification, Anti-*H. pylori*, anti-biofilm, anti-inflammatory, *S. aromaticum*, GC–MS main strategies

## Abstract

Eugenol essential oil (EEO) is the major component in aromatic extracts of *Syzygium aromaticum* (clove) and has several biological properties, such as antibacterial, antioxidant, and anti-inflammatory activities, as well as controlling vomiting, coughing, nausea, flatulence, diarrhea, dyspepsia, stomach distension, and gastrointestinal spasm pain. It also stimulates the nerves. Therefore, the aim of this study was to extract and purify EEO from clove buds and assess its ability to combat resistant *Helicobacter pylori*. Additionally, EEO’s anti-inflammatory activity and its ability to suppress *H. pylori* biofilm formation, which is responsible for antibiotic resistance, was also investigated. *Syzygium aromaticum* buds were purchased from a local market, ground, and the EEO was extracted by using hydro-distillation and then purified and chemically characterized using gas chromatography–mass spectrometry (GC–MS). A disk-diffusion assay showed that *Helicobacter pylori* is sensitive to EEO, with an inhibition zone ranging from 10 ± 06 to 22 ± 04 mm. The minimum inhibition concentration (MIC) of EEO ranged from 23.0 to 51.0 μg/mL against both *Helicobacter pylori* clinical isolates and standard strains. In addition, EEO showed antibiofilm activity at 25 µg/mL and 50 µg/mL against various *Helicobacter pylori* strains, with suppression percentages of 49.32% and 73.21%, respectively. The results obtained from the anti-inflammatory assay revealed that EEO possesses strong anti-inflammatory activity, with human erythrocyte hemolysis inhibition percentages of 53.04, 58.74, 61.07, and 63.64% at concentrations of 4, 8, 16, and 32 μg/L, respectively. GC–MS analysis revealed that EEO is a major component of *Syzygium aromaticum* when extracted with a hydro-distillation technique, which was confirmed by its purification using a chemical separation process. EEO exhibited antibacterial action against resistant *Helicobacter pylori* strains, as well as antibiofilm and anti-inflammatory activities, and is a promising natural alternative in clinical therapy.

## 1. Introduction

*Helicobacter pylori (H. pylori)* is an important pathogenic microbe in the gastrointestinal tract that colonizes the mucus layer of the stomach in about 50% of humans worldwide and causes chronic gastritis, peptic ulcers, and gastric cancer. The increased rate of *H. pylori* resistance to antibiotics has created a dangerous situation. The eradication of *H. pylori* is an important goal in combating gastric diseases. Several regimens are currently available, but none of them can accomplish the eradication of *H. pylori* and the inflammation it causes. For this reason, the search for alternative and effective new therapeutic methods is critical [1]. Natural products of plant origin have been used throughout history as therapeutic bioresources and are described in texts from many nations. In ancient civilizations, plant food was used for medical therapy. Using natural products in therapeutic management against microbial infections such as *H. pylori* has advantages over therapies from synthetic sources. They are often chosen because they have fewer side effects when their toxicological and pharmacological activities are compared to those derived from chemical sources [2]. The decreased toxicity of natural products has prompted interest in the pharmacological activities that these products have against *H. pylori*. In the last decade, various studies have assessed the use of plants and plant extracts and constituents as gastroprotective agents against *H. pylori* activity [3]. *Syzygium aromaticum*, known as clove, is from the family Myrtaceae and has been shown to possess several biological activities [4]. *S. aromaticum* is a rich source of bioactive substances and has several therapeutic properties, including the control of vomiting, nausea, cough, flatulence, dyspepsia, diarrhea, stomach distension, relieving gastrointestinal spasm pain, relieving uterine contractions, and possessing anti-inflammatory activity [1,5,6]. Eugenol, a phenolic compound, is the main constituent of *Syzygium aromaticum* extract. It has antioxidant activity [7], includes a monoamine oxidase inhibitor, and has neuroprotective activity [8]. Additionally, eugenol exhibits excellent bactericidal activity against a broad range of bacteria, such as *Staphylococcus aureus*, *Escherichia coli*, *H. pylori* [1,9] and *Listeria monocytogenes* [10]. Moreover, some studies have suggested that the mode of antibacterial action of eugenol is through the disruption of the cytoplasmic membrane by increasing its non-specific permeability. Additionally, the hydrophobic nature of eugenol molecules helps to penetrate the lipopolysaccharide in the cell wall of Gram-negative bacteria and changes their cell structure, resulting in the infiltration of intracellular constituents [1,11,12]. Burt [11] revealed that the hydroxyl group on eugenol is associated with proteins that inhibit enzyme action in the bacterial cell. The antibacterial activity of several natural products has been reported against *H. pylori*. Eugenol is a major component of methanolic extracts of *S. aromaticum* and exhibits highly significant antibacterial activity against resistant strains of *H. pylori* [13]. Furthermore, eugenol prevents the growth of all 30 *H. pylori* strains at a concentration of 2 μg/mL. Additionally, *H. pylori* did not develop any resistance toward eugenol even after 10 passages grown at sub-inhibitory concentrations [14]. We aimed to extract the eugenol from an *S. aromaticum* extract and investigate its antibacterial, antibiofilm and anti-inflammatory activities against resistant *H. pylori*.

## 2. Results

### 2.1. Antibacterial Activity of *EEO* and MIC

In the worldwide traditional medical system, several plants and plant products are known to possess potent medicinal advantages, suggesting that plants, plant products, and their extracts may be useful for specific medical cases. Hence, in our attempt to identify some active natural substances that have the ability to inhibit the growth of resistant *H. pylori*, we extracted EEO from *S. aromaticum* buds and assessed its ability to inhibit the growth of the bacterial strains under study. The results obtained from the extraction and purification process revealed a yield of 0.35% (*w*/*w*). The antibacterial activity of EEO assessed using the disc diffusion method showed that EEO displayed an inhibition zone diameter with mean ± SD (from 10 ± 06 to 22 ± 04 mm) against all isolates and the standard strain. These results were higher than that of amoxicillin (12.0, 12.0, 14.03, 16 ± 0.5) against the isolates of HPM 52, HPM65, HPM37, and standard strain NCTC 11637, respectively. The MIC of EEO ranged from 23.0 to 51 μg/mL against both. *H. pylori* isolates and standard strain NCTC 11637, as shown in Table 1 and Table 2 and Figure 1.

### 2.2. Biofilm Formation and Its Suppression by EEO

In this study, 76 *H. pylori* isolates were tested for their ability to form a biofilm. A total of 71 (93.36%) of the bacterial isolates formed biofilms with different degrees, the moderate degree of biofilm formed is reflected in the high frequency between *H. pylori* isolates under investigation. The bacterial strains formed a biofilm after the treatment with EEO at concentrations of 25 and 50 µg/mL with percentages of 36% (50.68) and 19% (26.75), respectively, as recorded in Table 3. Moreover, the results obtained from the antibiofilm activity of EEO showed suppression percentages of the formed biofilm with 49.32% and 73.21% at concentrations of 25 and 50 µg/mL, respectively. In contrast, amoxicillin yielded a suppression of biofilm formation with percentages of 43.46% and 92.37% at the same concentrations. The maximum suppression of biofilm formation by EEO was observed at 50 µg/mL, with (73.21%), as shown in Figure 2.

### 2.3. Development of Bacterial Resistance

In this study, we investigated the antibiotic resistance acquirement of *H. pylori* strains after exposure to subinhibitory concentrations of EEO. The antibiotic susceptibility of the bacterial isolates and standard NCTC 11637 strain before and after exposure to EEO several times revealed that antibiotic susceptibility had not changed, as shown in the results recorded in Table 4. This clarifies that the tested bacterial strains did not acquire resistance to antibiotics after exposure to EEO at the sub-inhibitory concentration (12 μg/mL) for 24 h several times.

### 2.4. Anti-Inflammatory Activity of EEO

Table 5 shows the inflammatory results obtained from the four different treatments with EEO and sodium diclofenac as a positive control. The purified EEO exhibited inhibition percentages of human erythrocyte hemolysis of 53.04%, 58.74%, 61.07%, and 63.64% at concentrations of 4, 8, 16, and 32 μg/L compared with inhibition by sodium diclofenac with 63.72%, 67.49%, 69.18%, and 71.43% at the same concentrations, respectively.

### 2.5. GC–MS Analysis of EEO

The GC–MS analysis of the methanolic extract of *Syzygium aromaticum* revealed that eugenol is the major component, with a percentage of 96.35%, while ethyl eugenol occupied 3.65%, which confirmed the extraction of EEO from *Syzygium aromaticum* buds with the hydro-distillation method. The purified EEO exhibited a maximum peak at 16.22, molecular formula C_10_H_12_O_2_ and a molecular weight of 164 *m/z*, as shown in Figure 3A,B.

## 3. Discussions

*Helicobacter pylori* is an important bacterial species in the gastrointestinal system that colonizes the mucoid layer of the stomach in about 50% of humans worldwide and causes several diseases, including chronic gastritis, peptic ulcers, and gastric cancer. The World Health Organization (WHO) recorded *H*. *pylori* as a class I carcinogen because of its role in cancer [1]. Currently, first-line treatment depends on the combination of two antibiotics, clarithromycin and amoxicillin or metronidazole with a proton pump inhibitor (triple therapy). Levofloxacin can be used as an alternative to clarithromycin in first-line therapy, with higher treatment rates. In recent years, with the rapidly increasing resistance of *H. pylori* to antibiotics, the treatment of this microbe has remained a major challenge to physicians [15]. For this reason, the eradication of *H. pylori* is very critical to patients worldwide. From this point, our study aims to research a new approach to eradicating resistant *H. pylori* and its virulence factors. In this context, we extracted EEO as an active compound from *S. aromaticum* buds to combat resistant *H. pylori.* In this study, the EEO extracted from *S. aromaticum* buds exhibited antibacterial activity against *H. pylori* isolates and standard strain NCTC 11637, with an inhibition zone diameter with a mean ± SD (from 10 ± 06 to 22 ± 04 mm), comparable to amoxicillin (from 0.0 to 16 ± 0.05). In addition, the MIC of EEO ranged from 23 to 51 μg/mL against both. *H. pylori* isolates and standard strain NCTC 11637. A previous study by Ali and his colleague revealed that the ability of eugenol inhibits the growth of thirty strains of *H. pylori* at a concentration of 2 μg/mL [14]. Moreover, *H. pylori* did not develop any resistance toward eugenol even after 10 passages grown at sub-inhibitory concentrations. Furthermore, El-Shouny et al. [13] reported that the methanolic extract of clove exhibited activity against resistant strains of *H. pylori*, with an inhibition zone ranging from 20 ± 0.57 to 25 ± 0.56 mm, where eugenol was detected as a major constituent, with a percentage (28.14%). Clove oil is recorded as a “generally regarded as safe compound” by the FDA when used at levels not exceeding 1500 ppm in food categories. In addition, the World Health Organization (WHO) and Expert Committee on Food Additives have established the acceptable daily human intake of clove oil at 2.5 mg/kg body weight for humans [16]. Several studies proposed that the mode of antibacterial action of eugenol is due to the disruption of the plasma membrane, which exceeds its non-specific permeability. Furthermore, the hydrophobic nature of eugenol enables it to permeate the lipopolysaccharide of the Gram-negative bacterial plasma membrane and modifies the cell structure, which then results in the infiltration of intracellular constituents [1,12]. The antibiofilm study revealed that EEO possesses strong antibiofilm activity against resistant *H. pylori* isolates with various degrees; the maximum suppression was observed (73.21%) at a concentration of 50 µg/mL. Previous studies have reported that eugenol can suppress and eradicate the biofilms caused by *Vibrio parahaemolyticus* [17] *Staphylococcus aureus* [18], *Candida albicans* [19], *Candida tropicalis*, *Candida. dubliniensis* [20], *Porphyromonas gingivalis* [21], *Aggregatibacter actinomycetemcomitans* ATCC 43718 [22], *Staphylococcus aureus* ATCC25923 [23], *Escherichia coli* O157:H7 [24], and *Streptococci* [25]. Additionally, significant biofilm reduction by clove extracts at various concentrations has been reported [26,27]. Our finding revealed that the sublethal dose of EEO did not develop bacterial resistance after exposure to EEO. These results are consistent with the results reported by Ali et al. [14]. *H. pylori* acquired resistance to antibiotics, such as clarithromycin and amoxicillin, after 10 sequential passages, as recorded previously [28]. In our study, we investigated the anti-inflammatory activity of EEO; the results obtained from the four treatments revealed the strong activity of EEO to inhibit human erythrocyte hemolysis with 53.04%, 58.74%, 61.07%, and 63.64% at concentrations of 4, 8, 16, and 32 μg/mL, respectively. The ability of drugs to stabilize erythrocyte membranes can also stabilize lysosomal membranes and, therefore, exhibit anti-inflammatory properties by changing the activity and release of cell mediators due to the resemblance between them [29]. Eugenol, as a natural compound with anti-inflammatory activity, has gained a great deal of attention in topical applications [30]. Capasso and his colleague reported eugenol to have both antiulcerogenic and anti-inflammatory activities [31]. In this context, eugenol was used before 1950 in the treatment of ulcer disease [32]. Previous studies have reported that eugenol has anti-inflammatory activities comparable with some NSAIDs, such as sodium diclofenac and indomethacin. Genetic studies of the anti-inflammatory activity of eugenol showed COX-2 inhibition without affecting COX-1 in mice macrophage cell cultures [33]. The GC–MS of methanolic extracts of *S. aromaticum* revealed that eugenol is a major compound, with a maximum peak of 16.4. The results obtained were consistent with DeFrancesco [34]. In Brazil, eugenol extracted from *S. aromaticum* essential oil was the predominate component, with a high percentage of (90.3%), in addition to eugenol acetate (1.87%) and β-caryophyllene (4.83%) [35]. *S. aromaticum* essential oil obtained in Italy [36] and China [21] also had eugenol as a major component, with 77.9% and 90.84%, respectively. These results agreed with the percentage obtained in the present study. The percentage of eugenol contained in the essential oil and the difference between the compounds can be directly related to the different geographic areas where the plant has been grown up, which can be affected or altered by abiotic and biotic factors such as the stage and age of plant growth, season, and climatic changes [37]. Moreover, the extraction method used to obtain the essential oil can also change its chemical constituents, as distillation and storage conditions can influence the content of its volatile oils. The change in chemical composition directly affects pharmacological and biological values, as noted in the anti-inflammatory and antimicrobial activities [38].

## 4. Materials and Methods

### 4.1. Plant Material

*S. aromaticum* was purchased from the local market of a Cairo governorate in Egypt. *S. aromaticum* was washed with distilled water, allowed to dry at room temperature, and powdered using an electric blender.

### 4.2. Extraction of EEO from S. aromaticum Buds

Eugenol was extracted from *S. aromaticum* buds using a hydro-distillation method with a Cleavenger-type apparatus for 2.5 h [39,40]. Approximately 300 g of powdered *S. aromaticum* was diluted in water in the proportion 1:10 (*S. aromaticum* + water) and extracted with the hydro-distillation method using a Clevenger system for 2.5 h at 100 °C. The EEO was collected and dried with anhydrous sodium sulfate (Na_2_SO_4_), and the final volume found was used to assess the yield through the mass/volume ratio by measuring the density. Mass/volume ratios were calculated from the mass (g) of the initial *S. aromaticum* material and the volume (mL) of obtained eugenol from the extraction process. The extracts were stored in sealed vials at 4 °C. for further studies.

### 4.3. Purification of Extracted EEO

An amount of 25 mL of EEO crude extract was transferred to a separating funnel, mixed with 50 mL of dichloromethane, and shaken well. The mixture was then allowed to separate into two distinct layers. In a 200 mL conical flask, the lower layer was collected, and 25 mL of dichloromethane was added a second time. The contents of the conical flask were transferred to a separating funnel, shaken well, and allowed to separate into two layers. The lower layer was collected in a conical flask and transferred to a clean separating funnel. Then, 40 mL of a 10% sodium hydroxide solution was added to the separating funnel to separate the eugenol acetate. The aqueous layer was collected in a conical flask, and the solution was acidified to a pH of less than 2.0 using conc. HCl. The pH was checked, and the aqueous layer was washed with a half-saturated sodium chloride solution. The organic phase was collected, and 25 mL of dichloromethane was added to it. The solution was dried with magnesium sulphate and filtered with filter paper. Finally, it was evaporated in a rotatory evaporator at a temperature of 60 °C until it became pure eugenol as a light-yellow oil [41]. The pure eugenol was characterized using GC–MS.

### 4.4. Anti-H. pylori Activity of EEO

The antibacterial activity of *eugenol* essential oil was determined against multidrug-resistant *H. pylori* isolates previously isolated from clinical samples and identified, as well as a standard strain of *H. pylori* NCTC 11637. Muller Hinton blood Agar (MHBA) medium was inoculated with 100 μL of bacterial growth (1.5 × 10^6^ CFU/mL). Paper discs (8 mm) were saturated with 50 µL of eugenol extracts at a concentration of 10 mg/mL(eugenol/DMSO). The saturated paper discs were placed on the surface of agar plates inoculated with bacterial test strains; the antibiotic amoxicillin 25 µg/mL was used as a positive control on the same plates. The plates were incubated at 37 °C for 48 h under microaerophilic conditions (10% CO_2_, 5% O_2_, and 85% N_2_) using a gas pack system (Mitsubishi, Japan), and the inhibition zone diameter was estimated in millimeters (mm). This experiment was performed in three replicates [42,43].

### 4.5. Determination of Minimum Inhibitory Concentrations (MICs) of EEO

The MIC of EEO against *H. pylori* isolates and *H. pylori* NCTC 11637 was performed by the microdilution broth method using amoxicillin (HiMedia Laboratories Pvt. Ltd., Thane, India) as a positive control in a 96-well microplate. Muller Hinton broth (MHB) medium was inoculated with a cell suspension of *H. pylori* isolates and *H. pylori* NCTC 11637 (10^6^ CFU/mL), and 100 µL of the inoculated medium was distributed in each well. The antibacterial substances (amoxicillin and EEO) were tested in a two-fold serial dilution, and the cultures were incubated at 35 °C for 3 days in a microaerophilic atmosphere (10% CO_2_, 5% O_2_, and 85% N_2_). EEO was investigated at concentrations of 0.0, 5.0, 10.0, 15.0, 20.0, 25.0, 30.0, 35.0, 40.0, 45.0, 50.0, 55.0, and 60.0 µg/mL and amoxicillin at 0.30, 0.613, 1.25, 2.5, 5.0, 10.0, 20.0, 40.0, and 60 µg/mL. The experiment was performed according to the criteria of (CLSI, 2020 guidelines (M7-A5) [42]. One of the negative control wells contained medium and EEO, while the other contained medium and amoxicillin at the tested concentrations, which were analyzed to determine the differences in optical density (O.D.) at 630 nm. MIC was defined as the lowest concentration of the EEO or amoxicillin, which can inhibit the visible growth of bacteria.

### 4.6. Antibiofilm Activity of EEO

A quantitative assessment of biofilm formation suppressed by EEO was performed using the micro-titer plate technique. Bacterial suspensions were prepared from 24-h-old cultures of each isolate. The turbidity of the initial suspension was adjusted by comparing it with a 0.5 McFarland standard. The initial bacterial suspensions contained approximately 10^8^ CFU/mL, which were then diluted to 1:100 by sterile 0.85% saline solution. Then, 100 μL of this dilute was inoculated, in triplicate, in a 96-well flat-bottomed polystyrene plate (China) and incubated for 24 h at 37 °C. The content of each well was discharged, and the wells were washed several times with phosphate-buffered saline (PBS). After that, methanol fixation was conducted for 15 min, and the plate was then air-dried. Each well was stained with 100 μL of 1% crystal violet solution in water and incubated at room temperature for 30 min. Afterward, the stain was solubilized by 100 μL of glacial acetic acid (GAA) 33%; the plates were washed with distilled water three times and then dried. The optical density (OD) of each well at 570 nm was read using an enzyme-linked immunosorbent assay (ELISA). The cut-off OD control for the microtiter plate was defined as three standard deviations (SD) plus the mean OD of the negative control. Based on the OD average values, the results of the biofilm formation were interpreted as follows; (OD ≤ ODC) = negative (non-biofilm formation) ODC = optical density of control; (ODC ≤ OD ≤ 2ODC) = weak biofilm formation, (2ODC ≤ OD ≤ 4ODC) = moderate- biofilm formation and (4ODC ≤ OD) = strong biofilm formation [28].

### 4.7. Development of Bacterial Resistance

Four *H. pylori* isolates and standard strain NCTC 11637 were used to evaluate the ability of bacterial strains to develop antibiotic resistance after exposure to EEO. The antibiotics susceptibility of these bacteria to three different antibiotics (clarithromycin, metronidazole and amoxicillin) were as follows: standard strain NCTC 11637 (sensitive to the three antibiotics), isolate HPM52 (sensitive to the three antibiotics), HPM56 (resistant to the three antibiotics), HPM37 (resistant to clarithromycin, metronidazole, and sensitive to amoxicillin), and HPM65 (resistant only to clarithromycin) and they were grown on agar at sub-inhibitory concentrations (12.0 μg/mL) of eugenol for 48 h. After the end of the incubation period, antibiotic susceptibility was retested again. This experiment was performed in three replicates [14].

### 4.8. Anti-Inflammatory Assay of EEO by Human RBCs

The anti-inflammatory activity of EEO was investigated using the human red blood cell membrane stabilization technique. The blood sample, collected from a healthy human volunteer who had not taken any NSAIDS 2 weeks prior to the experiment, was mixed with an equal volume of Alsever solution (2% dextrose, 0.8% sodium citrate, 0.5% citric acid and 0.42% NaCl), and centrifuged at 3000 rpm. The precipitated cells were washed with iso-saline and a 10% suspension was prepared. Different concentrations of EEO and sodium diclofenac were prepared viz., 4, 8, 16, and 32 μg/mL using dimethyl sulfoxide (DMSO). Then, 1 mL of phosphate buffer, 2 mL of hypo-saline, and 0.5 mL of human red blood cells (HRBC) suspension were added to the previous concentrations.

Then, they were incubated at 37 °C for 30 min and centrifuged at 3000 rpm for 20 min. The estimation of the inhibition of HRBC hemolysis was conducted through the estimation of supernatant hemoglobin content by using a spectrophotometer at 560 nm [44]. Percentage (%) inhibitions of hemolysis = (absorbance of control − absorbance of the sample)/(absorbance of control) “× 100”.

### 4.9. Identification of *Eugenol* Essential Oil by GC–MS

The eugenol essential oil compounds were analyzed and identified using GC–MS, as described by Zothanpuia et al. [45], with minor modifications. In brief, the EEO was dissolved in spectroscopy-grade methanol, and the GC–MS analysis was carried out using a Thermo Scientific trace GC1310-ISQ mass spectrometer (Austin, TX, USA) with a direct capillary column (length 30 m, thickness 0.25 µm, internal diameter 25 mm). The oven temperature was set to 50 °C for 5 min, then scaled up to 230 °C at a rate of 5 °C/min and maintained for 2 min; 1 μL of the sample was injected at 250 °C using helium as a carrier gas, divided at a ratio of 1:30. The mass spectrometer was set to scan from 40 to 1000 *m/z* in electron ionization (EI) mode at 200 °C and 70 eV. The observed compounds’ spectra were compared with the spectra of known compounds contained in the WILEY 09 (Wiley, New York, NY, USA) and NIST 11 libraries.

### 4.10. Statistical Analysis

The data were expressed as the mean ± SD value, which was calculated by using Minitab 18 software extended with a statistical package and Microsoft Excel 365.

## 5. Conclusions

The data obtained in this study show that EEO essential oil extracted from *S. aromaticum* possesses powerful antibacterial and antibiofilm activities against antibiotic-resistant *H. pylori* as well as anti-inflammatory activity. Therefore, we propose that EEO can be used to improve the treatment and eradication of *H. pylori*. In addition, the administration of eugenol in small amounts can protect against *H. pylori* biofilm-related infection.

## Figures and Tables

**Figure 1 molecules-28-02448-f001:**
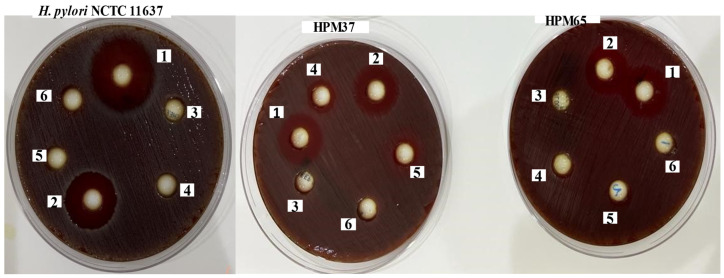
Antibacterial activity of purified eugenol (1), amoxicillin (2), water extract of *S. aromaticum* (3), Dimethyl sulfoxide (DMSO) (4), water (5) and mixed from water and DMSO (6) against resistant *H. pylori* isolates (HPM37 and HPM 65) and *H. pylori* NCTC 11637.

**Figure 2 molecules-28-02448-f002:**
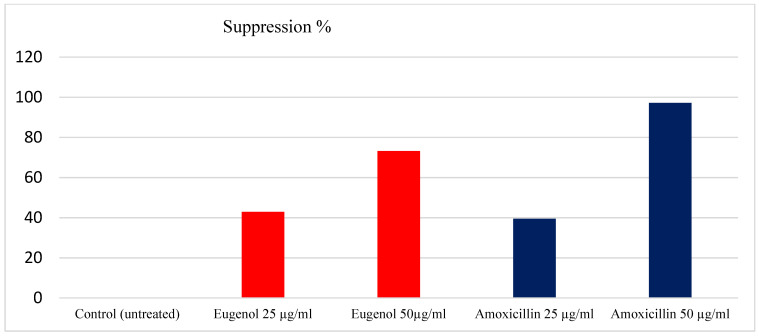
Percentage of biofilm suppression with EEO and amoxicillin.

**Figure 3 molecules-28-02448-f003:**
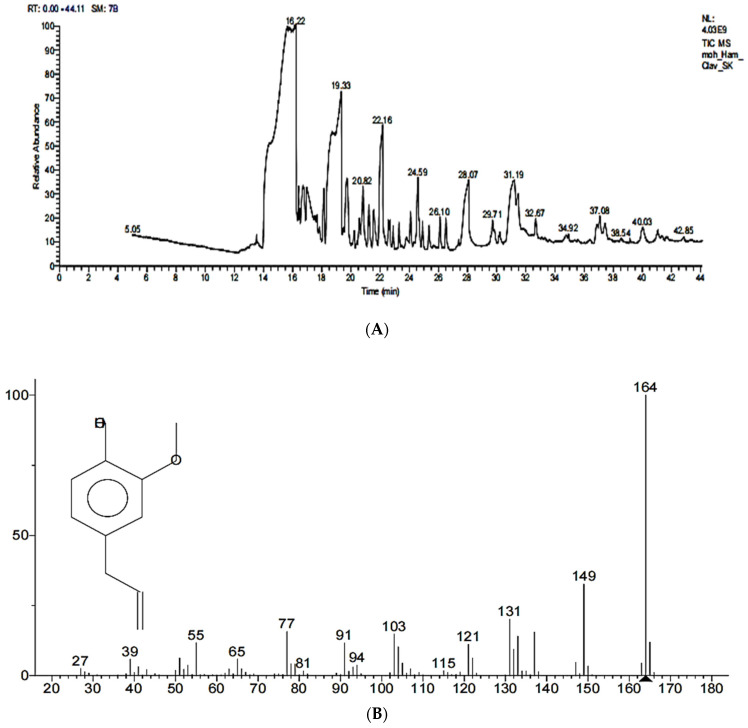
(**A**) GC–MS of methanolic extract of *Syzygium aromaticum.* (**B**) GC–MS of purified EEO in methanolic alcohol.

**Table 1 molecules-28-02448-t001:** Antibacterial activity of EEO against twenty-five resistant *H. pylori* isolates and standard strain.

No	Strains Code	Mean of Inhibition Zone Diameter mm (Mean ± SD)		No	Strains Code	Mean of Inhibition Zone Diameter mm (Mean ± SD)	
		EEO	Amoxicillin			EEO	Amoxicillin
1	HPM4	13 ± 03	0.0	14	HPM48	11 ± 15	0.0
2	HPM7	12 ± 04	0.0	15	HPM51	14 ± 10	0.0
3	HPM9	11 ± 07	0.0	16	HPM54	11 ± 10	0.0
4	HPM12	10 ± 15	0.0	17	HPM57	13 ± 20	0.0
5	HPM15	14 ± 06	0.0	18	HPM62	11 ± 00	0.0
6	HPM16	12 ± 00	0.0	19	HPM63	13 ± 00	0.0
7	HPM19	14 ± 10	0.0	20	HPM65	13 ± 00	12.0.
8	HPM26	11 ± 04	0.0	21	HPM66	11 ± 20	0.0
9	HPM37	15 ± 05	14.03	22	HPM70	10 ± 21	0.0
10	HPM44	10 ± 06	0.0	23	HPM72	11 ± 16	0.0
11	HPM48	14 ± 08	0.0	24	HPM73	13 ± 15	0.0
12	HPM52	11 ± 00	12.0	25	HPM75	14 ± 08	0.0
13	HPM56	13 ± 05	0.0	26	*H. pylori* NCTC 11637	22 ± 04	16 ± 05

**Table 2 molecules-28-02448-t002:** MIC of EEO against resistant *H. pylori* isolates and standard strain.

No	Strains Code	Minimum Inhibitory Concentration (μg/mL)		No	Strains Code	Minimum Inhibitory Concentration (μg/mL)	
		EEO	Amoxicillin			EEO	Amoxicillin
1	HPM4	27.0	28.0	14	HPM48	40.40	36.0
2	HPM7	29.0	30.0	15	HPM51	29.0	47.0
3	HPM9	23.0	35.0	16	HPM54	24.60	35.0
4	HPM12	24.0	27.0	17	HPM57	29.90	37.0
5	HPM15	30.50	34.0	18	HPM62	39.10	40.0
6	HPM16	33.60	42.0	19	HPM63	51.0	49.0
7	HPM19	24.0	31.0	20	HPM65	28.0	3.0
8	HPM26	24.40	29.0	21	HPM66	32.50	50.0
9	HPM37	35.0	2.0	22	HPM70	48.0	44.0
10	HPM44	27.50	30.0	23	HPM72	45.0	31.0
11	HPM48	33.0	36.0	24	HPM73	36.0	48.0
12	HPM52	39.0	2.0	25	HPM75	50.0	43.0
13	HPM56	46.20	37.0	26	*H. pylori* NCTC 11637	21.0	2.0

**Table 3 molecules-28-02448-t003:** Detection of biofilm among *H. pylori* isolates before and after treatment with EEO.

Treatment		Bacterial Biofilm Formation *N* (%) Total *N* = 76	Degree (%) *N* = 71		
			Strong *N* (%)	Moderate *N* (%)	Weak *N* (%)
**Control**	--------	71 (93.36)	17 (23.93)	43 (60.54)	11 (15.48)
**EEO**	25 µg/mL	36 (50.68)	8 (22.21)	11 (30.54)	17 (47.20)
	50 µg/mL	19 (26.75)	0.0	9 (47.36)	10 (52.63)
**Amoxicillin**	25 µg/mL	43 (56.54)	11 (25.57)	19 (44.17)	13 (30.22)
	50 µg/mL	2 (2.63)	0.0	2 (100%)	0.0

**Table 4 molecules-28-02448-t004:** Antibiotic susceptibility before (after) exposure to EEO.

Bacterial Strains	Antibiotics Susceptibility before Exposure to EEO			Antibiotics Susceptibility before Exposure to EEO		
	Clarithromycin	Metronidazole	Amoxicillin	Clarithromycin	Metronidazole	Amoxicillin
NCTC 11637	S	S	S	S	S	S
HPM52	S	S	S	S	S	S
HPM56	R	R	R	R	R	R
HPM637	R	R	S	R	R	S
HPM65	R	S	S	R	S	S

S = Sensitive; R = Resistance.

**Table 5 molecules-28-02448-t005:** Assessment of anti-inflammatory activity of EEO.

Treatment	Concentration (μg/mL)	Absorbance 560 nm	% Inhibition of Hemolysis
Control	0.0	1.246	0.0%
EEO	4	0.585	53.04%
	8	0.514	58.74%
	16	0.485	61.07%
	32	0.453	63.64%
Sodium diclofenac	4	0.452	63.72%
	8	0.405	67.49%
	16	0.384	69.18%
	32	0.357	71.43%

## Data Availability

Not applicable.

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
