# Peer review of "Antibacterial, Antibiofilm and Anti-Inflammatory Activities of Eugenol Clove Essential Oil against Resistant Helicobacter pylori"

_molecules, 2023, doi:10.3390/molecules28062448_

Round 1

Reviewer 1 Report

as enclosed document

Author Response

Dear / Editor

After greeting

We would like to express our sincere thanks to you for your kind consideration of our manuscript. We are also deeply grateful to the Editor and Reviewers for their remarks and constructive suggestions. We now have a detailed response to the reviewers comments, and a summary of updates to the manuscript prompted by your suggestion. We believe that reviewers have helped us improve our manuscript greatly, and hope that you find our manuscript now ready for publication.

-----------------------------------------------------------------------------------------

Reply to Reviewer #1

Comment:  1- Extensive editing of English language and style required of manuscript

Reply   Our manuscript was subjected to English grammar editing to improve grammar in the revised version and rewriting some sentences.

Comment:  2- Redefine it since your testing material is not eugenol alone, but an extract mixture?

Reply  Thank you very much.  The aims of our study extract eugenol by Hydro-distillation method and purification

Comment:  3- Do you have further purified the eugenol (C10H12O2) and identified it by GC- MS. If you did, list the protocol and more explanation.

Reply Thank you very much for this suggestion. We purified eugenol and added protocol in the materials methods section  

Comment:  4- Why the stander deviation (86) is larger than the mean (10). Explain it.

Reply Thank you very much for this suggestion. We recalculate stander deviation and  correct some errors in the calculate stander deviation.

Reviewer 2 Report

The manuscript entitled: Antibacterial, antibiofilm and anti-inflammatory activities of eugenol as essential oil of (clove) against resistant Helicobacter pylori, authors investigated the antimicrobial and anti-biofilm activity of eugenol essential oil of clove towards Helicobacter pylori. The manuscript needs to be really improved, mainly by detailing/enhancing. Specific comments are added in the attached file.

Author Response

  Dear / Editor

After greeting

We would like to express our sincere thanks to you for your kind consideration of our manuscript. We are also deeply grateful to the Editor and Reviewers for their remarks and constructive suggestions. We now have a detailed response to the reviewers comments, and a summary of updates to the manuscript prompted by your suggestion. We believe that reviewers have helped us improve our manuscript greatly, and hope that you find our manuscript now ready for publication.

-----------------------------------------------------------------------------------------

Reply to Reviewer #2

Comment:  1- correct title.

Reply Thank you very much for this suggestion.  We corrected title manuscript into Antibacterial, antibiofilm and anti-inflammatory activities of eugenol clove essential oil against resistant Helicobacter pylori

Comment:  2- corrected some words to written in italic form and font size.

Reply Thank you very much for this suggestion.  We corrected the revised manuscript.

Comment:  3- Reference number 19.

Reply Thank you very much.  We corrected the revised manuscript.

Round 2

Reviewer 1 Report

Review comment

Please use GC-MS uniformly

Rewrite the content in 2.3. Purification of extracted EEO.

Rewrite underline. Paper discs (8mm) were saturated with 50μL of eugenol extracts at a concentration of (10 mg of eugenol for every each one/ml of DMSO).

Rewrite underline. EEO was investigated at concentrations; 0.0 5.0, 10.0, 15.0, 20,0, 25.0, 30.0, 35.0, 40,0,45.0, 50.0, 55.0, 60.0, μg/ml, and amoxicillin started with 0.30, 0.613. 1.25, 2.5, 5, 10, 20, 40, and 60μg/mL. (significant figures ????)

Rewrite underline. Wells containing negative control (me-dium + EEO or amoxicillin at the tested concentrations) were performed to determine the differences in optical density (O.D.) at 630 nm.

Rewrite the sentence. In brief, 0.5 McFarland turbidity of H. pylori isolates was prepared from 24h cultures of each isolate previously to 100-fold dilution using TBS (Li-ofilchem, Italy).

Rewrite the sentence. Then, added to each concentration, 1 ml of phosphate buffer, 2 ml hypo-saline and 0.5 mL of human red blood cells (HRBC) suspension.

Rewrite underline. as described by (Zothanpuia et al [22] with minor modifications

The antibacterial activity of EEO was assessed by the disc diffusion method showing that EEO displayed a higher inhibition zone diameter with mean ± SD (10±86 to 16±75mm) comparable with amoxicillin (18±0.24 ??? mm) against standard strain. The figure number (10±86 to 16±75mm) in underline do not match the figure in the table 1.

The inhibition zone is not clear for HPM65, NCTC11637 (sample 3)in Figure 1. It need to re-photograph.

Rewrite underline.

the moderate degree is a predominate between among of these bacterial isolates

Rewrite underline.

Table 4. Antibiotics susceptibility of H. pylori strains before and after exposure to EEO.

Antibiotics susceptibility before (after) exposure to EEO

Author Response

 Dear/ Reviewer

Comment:  1- Please use GC-MS uniformly

Reply         Thank you very much. Our manuscript was subjected to revision and

                 correct GC-MS in all manuscript text.

Comment:  2- Rewrite the content in 2.3. Purification of extracted EEO.

Reply                Thank you very much.  We rewrite.

Comment:  3- Rewrite underline. Paper discs (8mm) were saturated with 50μL of

                     eugenol extracts at a concentration of (10 mg of eugenol for every

                        each one/ml of DMSO).

Reply                Thank you very much. We rewrite.

Comment:  4- Rewrite underline. EEO was investigated at concentrations; 0.0 5.0, 10.0, 15.0, 20,0, 25.0, 30.0, 35.0, 40,0,45.0, 50.0, 55.0, 60.0, μg/ml, and amoxicillin started with 0.30,  0.613. 1.25, 2.5, 5, 10, 20, 40, and 60μg/mL

Reply                Thank you very much.    We rewrite.

Comment:  5- Rewrite underline. Wells containing negative control (me-dium + EEO or  amoxicillin at the tested concentrations) were performed to determine the differences in optical density (O.D.) at 630 nm.

Reply                Thank you very much.    We rewrite.

Comment:  6- Rewrite the sentence. In brief, 0.5 McFarland turbidity of H. pylori isolates was prepared from 24h cultures of each isolate previously to 100-fold dilution using TBS (Li-  ofilchem, Italy).

Reply                Thank you very much.    We rewrite and correct.

Comment:  7- Rewrite the sentence. Then, added to each concentration, 1 ml of phosphate buffer, 2 ml hypo-saline and 0.5 mL of human red blood cells (HRBC)

                            suspension.

Reply                Thank you very much.    We rewrite and correct.

Comment:  8- Rewrite underline. as described by (Zothanpuia et al [22] with minor modifications

Reply                Thank you very much.    We rewrite and correct.

Comment:  9- The antibacterial activity of EEO was assessed by the disc diffusion method showing that EEO displayed a higher inhibition zone diameter with mean ± SD (10±86 to 16±75mm) comparable with amoxicillin (18±0.24 ??? mm) agains standard strain. The figure number(10±86 to 16±75mm) in underline do not match the figure in the table 1.

Reply                Thank you very much.    We rewrite and correct.

Comment:  10- The inhibition zone is not clear for HPM65, NCTC11637 (sample 3)in Figure 1.  It need to re-photograph.

Reply                Thank you very much. We improved the picture, and we think the inhibition  zone became clearer.

Comment:  11- Rewrite underline. the moderate degree is a predominate between among of these bacterial isolates

Reply                Thank you very much.    We rewrite and correct.

Comment:  12 Table 4. Antibiotics susceptibility of H. pylori strains before and after exposure to EEO.

Reply                Thank you very much.    We rewrite and correct.

Reviewer 2 Report

 Please, check the font size, it still looks different for different words and italic (It doesn't need to be italics). In some sections in the text, the names of bacteria and the Latin name of the plant used are not italicized, and also some bacterial names are incorrect, for example, Aggregatibacte actinomycetemcomitans. 

Please review the entire article again for the situations mentioned

Author Response

Dear / reviewer

- Please, check the font size, it still looks different for different words and italic (It

    doesn't need to be italics). In some sections in the text, the names of bacteria

  and the Latin name of the plant used are not italicized, and also some bacterial

    names are incorrect, for example, Aggregatibacte actinomycetemcomitans. 

    Please review the entire article again for the situations mentioned

Reply            Thank you very much.  We check the revised manuscript.
